# On Conditional Tsallis Entropy

**DOI:** 10.3390/e23111427

**Published:** 2021-10-29

**Authors:** Andreia Teixeira, André Souto, Luís Antunes

**Affiliations:** 1CINTESIS—Centre for Health Technology and Services Research, Faculty of Medicine, University of Porto, 4200-450 Porto, Portugal; andreiasofiat@med.up.pt; 2MEDCIDS—Department of Community Medicine, Information and Decision in Health, Faculty of Medicine, University of Porto, 4200-450 Porto, Portugal; 3ADiT-LAB, Instituto Politécnico de Viana do Castelo, Rua Escola Industrial e Comercial Nun’Álvares, 4900-347 Viana do Castelo, Portugal; 4LASIGE, Faculdade de Ciências da Universidade de Lisboa, Campo Grande, 1749-016 Lisboa, Portugal; 5Departamento de Informática, Faculdade de Ciências da Universidade de Lisboa, Campo Grande, 1749-016 Lisboa, Portugal; 6Instituto de Telecomunicações, Av. Rovisco Pais, n 1, 1049-001 Lisboa, Portugal; 7Computer Science Department, Faculty of Sciences, University of Porto, Rua do Campo Alegre, 4169-007 Porto, Portugal; lfa@dcc.fc.up.pt

**Keywords:** Tsallis entropy, conditional Tsallis entropies, Generalizations of Shannon entropy

## Abstract

There is no generally accepted definition for conditional Tsallis entropy. The standard definition of (unconditional) Tsallis entropy depends on a parameter α that converges to the Shannon entropy as α approaches 1. In this paper, we describe three proposed definitions of conditional Tsallis entropy suggested in the literature—their properties are studied and their values, as a function of α, are compared. We also consider another natural proposal for conditional Tsallis entropy and compare it with the existing ones. Lastly, we present an online tool to compute the four conditional Tsallis entropies, given the probability distributions and the value of the parameter α.

## 1. Introduction

Tsallis entropy [1], (The name Tsallis entropy used in this paper, to identify the quantity presented in Equation (Equation 3), is not consensual in the community, given that before Tsallis presented it in 1988, and as he himself acknowledges, other authors had already introduced it [2,3,4].) a generalization of Shannon entropy [5,6], was extensively studied by Constantino Tsallis in 1988, and provides an alternative way of dealing with several characteristics of nonextensive physical systems, given that the information about the intrinsic fluctuations in the physical system can be characterized by the nonextensivity parameter α. It can be applied to many scientific fields, such as physics [7], economics [8], computer science [9,10], and biology [11]. We refer the reader to Reference [12] for a more extensive bibliography on applications of Tsallis entropy. Furthermore, we refer the reader to Reference [13] for a survey on the most significant areas of application of the most usual entropy measures, including Shannon [6], Rényi [14], and Tsallis entropies [1,2,3,4].

It is known that, as the parameter α approaches 1, the Tsallis entropy corresponds to the Shannon entropy. Unlike for Shannon entropy, but similar to Rényi entropy (yet another generalization of Shannon entropy developed by Alfréd Rényi in 1961 [14], which also depends on a parameter α and converges to Shannon entropy when α approaches 1), there is no commonly accepted definition for the conditional Tsallis entropy: several versions have been proposed and used in the literature [15,16]. In this work, we revisit the notion of conditional Tsallis entropy by studying some natural and desirable properties in the existing proposals (see for instance References [15,16]): when α→1, the usual conditional Shannon entropy should be recovered, the conditional Tsallis entropy should not exceed the unconditional Tsallis entropy, and the conditional Tsallis entropy should have values between 0 and the maximum value of the unconditional version.

The use of entropies in different fields, especially in the field of information theory and its connection to communication, allowed the development of several useful information measures, such as mutual information, symmetry of information, and information distances. See, for example, References [17,18,19] for some recent work related to the aforementioned information measures.

Depending on the entropy measure used, all of these have been applied in many different areas of knowledge, such as physics [20], information theory [21,22], complexity theory [23,24,25], security [26,27,28], biology [29,30,31,32], finances [33], and medicine [34,35,36], among others. The conditional Tsallis entropy, as suggested in Reference [37], can be directly applied to information theory, especially coding theory. Furthermore, since Tsallis entropy can be applied in many areas (see, for example, Reference [12]), the study of conditional Tsallis entropies is quite promising. This paper analyzes several definitions of conditional Tsallis entropy, with the intent of providing the reader with a description of the properties that each approach satisfies.

Continuing from previous works [37,38], we introduce a new natural definition for conditional Tsallis entropy as a possible alternative to the existing ones. Our new proposal does not intend to be the ultimate version of conditional Tsallis entropy, but an alternative to the existing ones, with its own properties that, in settings, such as biomedical applications, might be useful for defining information distances or other significant measurements. None of the known definitions contain all of the desired properties for a conditional version. In particular, the one presented here (as it takes the maximum over the marginal distributions) does not converge to the Shannon entropy when α→1—it behaves similar to a parameterized entropy, and is akin to the one proposed in Reference [38] as an alternative to Rényi’s conditional entropy, another generalization of Shannon entropy.

The paper is organized as follows. In the next section, we present the definitions necessary for the rest of the paper, namely Shannon entropy and Tsallis entropy. In Section 3, we provide several definitions for the conditional Tsallis entropy in both existing literature and our proposal. In Section 4, we establish several results, comparing the definitions presented previously. In Section 5, we explore some features of each variant for the conditional Tsallis entropy. Finally, in Section 6, we present the conclusions and future work.

## 2. Preliminaries

In the remainder of the paper, we use the standard notation for entropies and for probability distributions according to Reference [5]. For the sake of simplicity of notation, we use the notation log for the logarithm in base 2. We call the reader’s attention to the fact that, whenever we say that one entropy converges to another, it is always up to logarithmic factor that depends only on the choice of cardinality of the alphabet.

The Shannon entropy of *X* is the expectation of the surprise of an occurrence,
(1)$$H\left(X\right)=-\sum_{x}\left(P(X=x)\log P(X=x)\right).$$
The conditional Shannon entropy, H(Y|X), is the expectation over *x* of the entropy of the distribution P(Y|X=x),
(2)$$\begin{array}{ccc} H(Y|X) & = & E_{x,y}\left[\log\left(\frac{1}{P(Y=y|X=x)}\right)\right]. \end{array}$$
It is easy to derive the *chain rule* H(X,Y)=H(X)+H(Y|X): to get the average information contained in (X,Y), we may first get the average information contained in *X*, and add to it the average information of *Y*, given *X*.

The Tsallis entropy [1] was firstly introduced in [2,3] and is defined for a random variable *X* by:(3)$$T_{\alpha }\left(X\right)=\frac{1}{\alpha -1}\left(1-\sum_{x}P{\left(X=x\right)}^{\alpha }\right),(\mathrm{for}\;\alpha >0,\;\alpha \neq 1).$$

It is straightforward to show that, when the parameter α converges to 1, the value of the entropy converges to the Shannon entropy.

## 3. Conditional Tsallis Entropy: Four Definitions

We consider three definitions for conditional Tsallis entropy that already exist in the literature and introduce a new proposal. All definitions consider a positive parameter α.

**Definition** **1.**
*Let Z=(X,Y) be a random vector. One can define the following variants of conditional Tsallis entropy:*
*1.* 
*Definition of $T_{\alpha }\left(Y|X\right)$ from Reference [15]*

(4)
$$\begin{array}{ccc} T_{\alpha }\left(Y|X\right)\; & = & \;\sum_{x}P{\left(X=x\right)}^{\alpha }T_{\alpha }\left(Y|x\right) \end{array}$$


(5)
$$\begin{array}{ccc} \; & = & \;\frac{1}{\alpha -1}\sum_{x}P{\left(X=x\right)}^{\alpha }\left(1-\sum_{y}P{\left(Y=y|X=x\right)}^{\alpha }\right)\;\;. \end{array}$$


*One can easily verify that $T_{\alpha }\left(X,Y\right)=T_{\alpha }\left(Y|X\right)+T_{\alpha }\left(X\right)$ and, therefore, it satisfies the chain rule.*
*2.* 
*Definition of $S_{\alpha }\left(Y|X\right)$ from [16] (Definition 2.8)*

(6)
$$\begin{array}{ccc} S_{\alpha }\left(Y|X\right) & = & \sum_{x}P\left(X=x\right)T_{\alpha }\left(Y|X=x\right) \end{array}$$


(7)
$$\begin{array}{ccc} & = & \sum_{x}P\left(X=x\right)\frac{1}{\alpha -1}\left(1-\sum_{y}P{\left(Y=y|X=x\right)}^{\alpha }\right) \end{array}$$


(8)
$$\begin{array}{ccc} & = & \frac{1}{\alpha -1}\sum_{x}P\left(X=x\right)\left(1-\sum_{y}P{\left(Y=y|X=x\right)}^{\alpha }\right). \end{array}$$

*3.* 
*Definition of $S_{\alpha }^{\prime }\left(Y|X\right)$ from [16] (Definition 2.10)*

(9)
$$S_{\alpha }^{\prime }\left(Y|X\right)=\frac{1}{\alpha -1}\left(\frac{1-\sum_{x,y}P{\left(X=x,Y=y\right)}^{\alpha }}{\sum_{x}P{\left(X=x\right)}^{\alpha }}\right).$$




The first definition presented proposes that the conditional Tsallis entropy should be weighed by the probability of sampling X=x with parameter α, while the second one proposes that one uniformly weighs only the probability of sampling X=x. Therefore, notice that for the first definition presented, the value of α largely affects the value of the conditional Tsallis entropy. The idea for the third proposal is to distribute evenly the influence of the parameter α by the entire joint distribution.

Next, we present another possible definition of the conditional Tsallis entropy. This definition is based on Definition III.6 of [38] and captures the intuitive notion of defining the conditional entropy, by taking the maximum over all possible marginal distributions. Note that this definition is analogous to an existing one for the Rényi entropy; however, as we will show later, this proposal does not satisfy some of the expected basic properties.

**Definition** **2**(Definition of $T_{\alpha }^{\prime }\left(Y|X\right)$).
(10)$$T_{\alpha }^{\prime }\left(Y|X\right)=\frac{1}{\alpha -1}\max_{x}\left(1-\sum_{y}P{\left(Y=y|X=x\right)}^{\alpha }\right).$$

We opted to use different notations for the variants of the conditional Tsallis entropy in the last definition, to better distinguish them in the rest of the paper. In particular, we follow the same approach as in Reference [38].

The following expressions will be useful later.

**Theorem** **1.**
*Let Z=(X,Y) be a random vector. The following identities are true:*

(11)
$$\begin{array}{ccc} T_{\alpha }^{\prime }\left(Y|X\right) & = & \max_{x}T_{\alpha }\left(Y|X=x\right)\;(for\;\alpha >1) \end{array}$$


(12)
$$\begin{array}{ccc} T_{\alpha }^{\prime }\left(Y|X\right) & = & \min_{x}T_{\alpha }\left(Y|X=x\right)\;(for\;\alpha <1). \end{array}$$



## 4. Comparison of the Definitions

We now compare the above four definitions of the conditional Tsallis entropy by comparing whether or not the definition satisfies some common properties of an entropy measure. In the next theorem, we report two simple facts with straightforward proofs. We leave the details for the interested reader to check.

**Theorem** **2.**
*For any fixed joint probability distribution P(X,Y),*
*(i)* 
*$T_{\alpha }\left(Y|X\right)$, $S_{\alpha }\left(Y|X\right)$ and $S_{\alpha }^{\prime }\left(Y|X\right)$, as functions of α, are continuous and differentiable;*
*(ii)* 
*$T_{\alpha }^{\prime }\left(Y|X\right)$, as a function of α, is continuous for all α≠1.*



The following results provide the possible comparisons (in terms of values) between the proposed definitions. For the sake of organization, we split the comparison by types of entropy.

First we compare $T_{\alpha }\left(Y|X\right)$ with $S_{\alpha }\left(Y|X\right)$.

**Theorem** **3.**
*For all joint probability distributions P(X,Y) and for every α>0,*

(13)
$$\begin{array}{ccc} & & \;if\;\alpha <1:\;\;S_{\alpha }\left(Y|X\right)\leq T_{\alpha }\left(Y|X\right) \end{array}$$


(14)
$$\begin{array}{ccc} & & \;if\;\alpha =1:\;\;S_{\alpha }\left(Y|X\right)=T_{\alpha }\left(Y|X\right)=H\left(Y|X\right) \end{array}$$


(15)
$$\begin{array}{ccc} & & \;if\;\alpha >1:\;\;S_{\alpha }\left(Y|X\right)\geq T_{\alpha }\left(Y|X\right). \end{array}$$



**Proof.** Consider first the case α<1. In this case, we have that $P{\left(X=x\right)}^{\alpha }\geq P\left(X=x\right)$. Thus,
$$\begin{array}{ccc} P{\left(X=x\right)}^{\alpha }\times T_{\alpha }\left(Y|X=x\right) & \geq & P\left(X=x\right)\times T_{\alpha }\left(Y|X=x\right)\\ \Leftrightarrow \sum_{x}P{\left(X=x\right)}^{\alpha }\times T_{\alpha }\left(Y|X=x\right) & \geq & \sum_{x}P\left(X=x\right)\times T_{\alpha }\left(Y|X=x\right)\\ \Leftrightarrow T_{\alpha }\left(Y|X\right) & \geq & S_{\alpha }\left(Y|X\right). \end{array}$$For the case α=1, see the proof of Theorem 8.The case α>1 is similar to the previous one, but this time, the conclusion follows, since for α>1, $P{\left(X=x\right)}^{\alpha }\leq P\left(X=x\right)$. □

In the next theorem we provide the comparison between $T_{\alpha }^{\prime }\left(Y|X\right)$ and $S_{\alpha }\left(Y|X\right)$.

**Theorem** **4.**
*For all joint probability distributions P(X,Y) and for every α>0,*

(16)
$$\begin{array}{ccc} & & \;if\;\alpha \leq 1:\;\;T_{\alpha }^{\prime }\left(Y|X\right)\leq S_{\alpha }\left(Y|X\right) \end{array}$$


(17)
$$\begin{array}{ccc} & & \;if\;\alpha >1:\;\;T_{\alpha }^{\prime }\left(Y|X\right)\geq S_{\alpha }\left(Y|X\right). \end{array}$$



**Proof.** Consider first the case α<1. In this case, we have that $T_{\alpha }^{\prime }\left(Y|X\right)=\min_{x}T_{\alpha }\left(Y|X=x\right)$. So,
(18)$$\begin{array}{ccc} S_{\alpha }\left(Y|X\right) & = & \sum_{x}\left(P\left(X=x\right)\cdot T_{\alpha }\left(Y|X=x\right)\right) \end{array}$$
(19)$$\begin{array}{ccc} & \geq & \sum_{x}(P\left(X=x\right)\cdot \left[\min_{x}T_{\alpha }\left(Y|X=x\right)\right]) \end{array}$$
(20)$$\begin{array}{ccc} & = & \left(\min_{x}T_{\alpha }\left(Y|X=x\right)\right)\cdot \sum_{x}P\left(X=x\right) \end{array}$$
(21)$$\begin{array}{ccc} & = & \min_{x}T_{\alpha }\left(Y|X=x\right) \end{array}$$
(22)$$\begin{array}{ccc} & = & T_{\alpha }^{\prime }\left(Y|X\right). \end{array}$$The proof of the case α>1 is similar to the previous one but this time, the conclusion follows from the fact that, for α>1, $T_{\alpha }^{\prime }\left(Y|X\right)=\max_{x}T_{\alpha }\left(Y|X=x\right)$. □

As a consequence of the two previous results and the definitions, we can derive the relation between $T_{\alpha }^{\prime }$ and $T_{\alpha }$.

**Corollary** **1.**
*For all joint probability distributions P(X,Y) and for every α>0,*

(23)
$$\begin{array}{ccc} & & \;if\;\alpha \leq 1:\;\;T_{\alpha }^{\prime }\left(Y|X\right)\leq T_{\alpha }\left(Y|X\right) \end{array}$$


(24)
$$\begin{array}{ccc} & & \;if\;\alpha >1:\;\;T_{\alpha }^{\prime }\left(Y|X\right)\geq T_{\alpha }\left(Y|X\right). \end{array}$$



The proof follows directly from Theorems 3 and 4. Now, we derive the relation between $S_{\alpha }^{\prime }\left(Y|X\right)$ and $T_{\alpha }^{\prime }\left(Y|X\right)$.

**Theorem** **5.**
*For all joint probability distributions P(X,Y) and for every α>0,*

(25)
$$\begin{array}{ccc} & & \;if\;\alpha \leq 1:\;\;S_{\alpha }^{\prime }\left(Y|X\right)\geq T_{\alpha }^{\prime }\left(Y|X\right) \end{array}$$


(26)
$$\begin{array}{ccc} & & \;if\;\alpha >1:\;\;S_{\alpha }^{\prime }\left(Y|X\right)\leq T_{\alpha }^{\prime }\left(Y|X\right). \end{array}$$



**Proof.** Consider first the case α<1. Proving that $S_{\alpha }^{\prime }\left(Y|X\right)\geq T_{\alpha }^{\prime }\left(Y|X\right)$, by definition, is the same to prove:
(27)$$\begin{array}{c} \frac{\left(1-\frac{\sum_{x,y}P{\left(X=x,Y=y\right)}^{\alpha }}{\sum_{x}P{\left(x\right)}^{\alpha }}\right)}{\alpha -1}\geq \frac{\max_{x}\left(1-\sum_{y}P{\left(y|x\right)}^{\alpha }\right)}{\alpha -1}. \end{array}$$As α<1, we have that $\frac{1}{\alpha -1}<0$. Thus, proving Equation (Equation 27) is the same, proves that:
$$\begin{array}{c} 1-\frac{\sum_{x,y}P{\left(X=x,Y=y\right)}^{\alpha }}{\sum_{x}P{\left(X=x\right)}^{\alpha }}\leq \max_{x}\left(1-\sum_{y}P{\left(Y=y|X=x\right)}^{\alpha }\right)\\ \Leftrightarrow \frac{\sum_{x,y}P{\left(X=x,Y=y\right)}^{\alpha }}{\sum_{x}P{\left(X=x\right)}^{\alpha }}\geq \min_{x}\sum_{y}P{\left(Y=y|X=x\right)}^{\alpha }\\ \Leftrightarrow \sum_{x,y}P{\left(X=x,Y=y\right)}^{\alpha }\geq \left(\sum_{x}P{\left(X=x\right)}^{\alpha }\right)\times \min_{x}\left(\sum_{y}P{\left(Y=y|X=x\right)}^{\alpha }\right). \end{array}$$
Now, the result follows by observing that the last inequality is true, since, for α<1 and for every *x*, we have that
$$\min_{x}\left(\sum_{y}P{\left(Y=y|X=x\right)}^{\alpha }\right)\leq \sum_{y}P{\left(Y=y,X=x\right)}^{\alpha }.$$
The case α>1 is proved in a similar manner. □

Now, we derive the relation between $T_{\alpha }\left(Y|X\right)$ and $S_{\alpha }^{\prime }\left(Y|X\right)$.

**Theorem** **6.**
*For all joint probability distributions P(X,Y) and for every α>0,*

(28)
$$\begin{array}{ccc} & & \;if\;\alpha \leq 1:\;\;T_{\alpha }\left(Y|X\right)\geq S_{\alpha }^{\prime }\left(Y|X\right) \end{array}$$


(29)
$$\begin{array}{ccc} & & \;if\;\alpha >1:\;\;T_{\alpha }\left(Y|X\right)\leq S_{\alpha }^{\prime }\left(Y|X\right). \end{array}$$



**Proof.** Consider first the case α<1. Thus,
$$\begin{array}{c} T_{\alpha }\left(Y|X\right)\geq S_{\alpha }^{\prime }\left(Y|X\right)\\ \Leftrightarrow \frac{1}{\alpha -1}\sum_{x}\left(P{\left(X=x\right)}^{\alpha }\left(1-\sum_{y}P{\left(Y=y|X=x\right)}^{\alpha }\right)\right)\geq \frac{1}{\alpha -1}\left(1-\frac{\sum_{x,y}P{\left(X=x,Y=y\right)}^{\alpha }}{\sum_{x}P{\left(X=x\right)}^{\alpha }}\right) \end{array}$$
$$\begin{array}{c} \Leftrightarrow \sum_{x}\left(P{\left(X=x\right)}^{\alpha }\left(1-\sum_{y}P{\left(Y=y|X=x\right)}^{\alpha }\right)\right)\leq 1-\frac{\sum_{x,y}P{\left(X=x,Y=y\right)}^{\alpha }}{\sum_{x}P{\left(X=x\right)}^{\alpha }}\\ \Leftrightarrow \sum_{x}P{\left(X=x\right)}^{\alpha }-\sum_{x}\left(P{\left(X=x\right)}^{\alpha }\sum_{y}\frac{P{\left(X=x,Y=y\right)}^{\alpha }}{P{\left(X=x\right)}^{\alpha }}\right)\leq 1-\frac{\sum_{x,y}P{\left(X=x,Y=y\right)}^{\alpha }}{\sum_{x}P{\left(X=x\right)}^{\alpha }}\\ \Leftrightarrow \sum_{x}P{\left(X=x\right)}^{\alpha }-\sum_{x,y}P{\left(X=x,Y=y\right)}^{\alpha }\leq \frac{\sum_{x}P{\left(X=x\right)}^{\alpha }-\sum_{x,y}P{\left(X=x,Y=y\right)}^{\alpha }}{\sum_{x}P{\left(X=x\right)}^{\alpha }}. \end{array}$$
The result follows by observing that the last inequality is true, since for α<1, we have that:
(30)$$\begin{array}{c} P{\left(X=x\right)}^{\alpha }>P\left(X=x\right) \end{array}$$
and consequently,
(31)$$\begin{array}{c} \sum_{x}P{\left(X=x\right)}^{\alpha }>1. \end{array}$$
The proof of the case α>1 is similar to the previous one. □

Finally, we show that the values of $S_{\alpha }$ and $S_{\alpha }^{\prime }$ are incomparable in the sense that there are probability distributions for which $S_{\alpha }$ is greater than $S_{\alpha }^{\prime }$ and there are probability distributions for which $S_{\alpha }^{\prime }$ is greater than $S_{\alpha }$.

**Theorem** **7.**
*The values of $S_{\alpha }\left(Y|X\right)$ and of $S_{\alpha }^{\prime }\left(Y|X\right)$ are incomparable, i.e., for each n≥2 and α≠1*

(32)
$$\begin{array}{c} \exists P\left(X,Y\right):\;\;S_{\alpha }\left(Y|X\right)<S_{\alpha }^{\prime }\left(Y|X\right) \end{array}$$


(33)
$$\begin{array}{c} \exists P\left(X,Y\right):\;\;S_{\alpha }\left(Y|X\right)>S_{\alpha }^{\prime }\left(Y|X\right). \end{array}$$



**Proof.** For Statement (Equation 32) and α<1, consider the following joint probability distribution:
$$\begin{array}{ccc} X\backslash Y & 1 & 2\\ 1 & 0.0625 & 0.0625\\ 2 & 0.0125 & 0.8625 \end{array}$$
(34)$$\begin{array}{ccc} S_{0.25}\left(Y|X\right) & \approx & 0.513 \end{array}$$
(35)$$\begin{array}{ccc} S_{0.25}^{\prime }\left(Y|X\right) & \approx & 0.629 \end{array}$$
For Statement (Equation 32) and α>1, consider the following joint probability distribution:
$$\begin{array}{ccc} X\backslash Y & 1 & 2\\ 1 & 0.1125 & 0.0125\\ 2 & 0.4375 & 0.4375 \end{array}$$
(36)$$\begin{array}{ccc} S_{2.5}\left(Y|X\right) & \approx & 0.396 \end{array}$$
(37)$$\begin{array}{ccc} S_{2.5}^{\prime }\left(Y|X\right) & \approx & 0.429 \end{array}$$
For Statement (Equation 33) and α<1, consider the following joint probability distribution:
$$\begin{array}{ccc} X\backslash Y & 1 & 2\\ 1 & 0.125 & 0\\ 2 & 0.5 & 0.375 \end{array}$$
(38)$$\begin{array}{ccc} S_{0.25}\left(Y|X\right) & \approx & 0.792 \end{array}$$
(39)$$\begin{array}{ccc} S_{0.25}^{\prime }\left(Y|X\right) & \approx & 0.560 \end{array}$$
Finally, for Statement (Equation 33) and α>1, consider the following joint probability distribution:
$$\begin{array}{ccc} X\backslash Y & 1 & 2\\ 1 & 0.0625 & 0.0625\\ 2 & 0.0125 & 0.8625 \end{array}$$
(40)$$\begin{array}{ccc} S_{1.25}\left(Y|X\right) & \approx & 0.125 \end{array}$$
(41)$$\begin{array}{ccc} S_{1.25}^{\prime }\left(Y|X\right) & \approx & 0.099 \end{array}$$
□

## 5. Properties of the Conditional Tsallis Entropies

In this section, we investigate some properties of the proposals considered. In particular, we show that there are probability distributions and α≠1 for which the conditional Tsallis entropies are bigger than the unconditional Tsallis entropy.

**Theorem** **8.**
*For any fixed joint probability distribution P(X,Y),*

(42)
$$\begin{array}{ccc} \lim_{\alpha \to 1}T_{\alpha }\left(Y|X\right) & = & H(Y|X) \end{array}$$


(43)
$$\begin{array}{ccc} \lim_{\alpha \to 1}S_{\alpha }\left(Y|X\right) & = & H(Y|X) \end{array}$$


(44)
$$\begin{array}{ccc} \lim_{\alpha \to 1}S_{\alpha }^{\prime }\left(Y|X\right) & = & H(Y|X) \end{array}$$

*where H(Y|X) is the conditional Shannon entropy. In general, it is not true that $\lim_{\alpha \to 1}T_{\alpha }^{\prime }\left(Y|X\right)=H\left(Y|X\right)$.*


**Proof.** The second equation is easy to derive directly from the definition of conditional probability and from Equation (Equation 2). Furthermore, using Equation (Equation 6) we can also easily obtain (using the previous derivation) that Equation (Equation 42) is also true.The third equation was proven in Reference [16].Now, it is only left to prove the last statement of the theorem, i.e., in general
(45)$$\begin{array}{c} \lim_{\alpha \to 1}T_{\alpha }^{\prime }\left(Y|X\right)\neq H\left(Y|X\right). \end{array}$$From Equations (Equation 6) and (Equation 11) it is easy to check that $T_{\alpha }\left(Y|X\right)$ is the *expectation* over *x* of $T_{\alpha }\left(Y|x\right)$, while $T_{\alpha }^{\prime }\left(Y|X\right)$ is the *maximum* over *x* of $T_{\alpha }\left(Y|x\right)$.The function $T_{\alpha }\left(Y|x\right)$ depends on the conditional probabilities P(Y=y|X=x). Therefore, there are joint probability distributions P(X=x,Y=y), such that:
(46)$$\begin{array}{c} \lim_{\alpha \to 1}T_{\alpha }^{\prime }\left(Y|X\right)\neq \lim_{\alpha \to 1}T_{\alpha }\left(Y|X\right)=H\left(Y|X\right). \end{array}$$
□

Contrary to the Shannon entropy, the value of any conditional Tsallis entropy may exceed the corresponding unconditional Tsallis entropy for all proposals.

**Theorem** **9.**
*There are probability distributions P(X,Y) and values of α, such that:*

(47)
$$\begin{array}{c} T_{\alpha }\left(Y|X\right)>T_{\alpha }\left(Y\right) \end{array}$$


(48)
$$\begin{array}{c} S_{\alpha }\left(Y|X\right)>T_{\alpha }\left(Y\right) \end{array}$$


(49)
$$\begin{array}{c} S_{\alpha }^{\prime }\left(Y|X\right)>T_{\alpha }\left(Y\right) \end{array}$$


(50)
$$\begin{array}{c} T_{\alpha }^{\prime }\left(Y|X\right)>T_{\alpha }\left(Y\right). \end{array}$$



**Proof.** Consider the following joint probability distribution:
$$\begin{array}{ccc} X=x\backslash Y=y & 1 & 2\\ 1 & 0.45 & 0.45\\ 2 & 0.1 & 0.0 \end{array}$$
For this distribution we have:
(51)$$\begin{array}{ccc} T_{0.5}\left(Y\right) & \approx & 0.824 \end{array}$$
(52)$$\begin{array}{ccc} T_{0.5}\left(Y|X\right) & \approx & 1.047 \end{array}$$
(53)$$\begin{array}{ccc} S_{0.5}\left(Y|X\right) & \approx & 0.828 \end{array}$$
(54)$$\begin{array}{ccc} T_{3}\left(Y\right) & \approx & 0.371 \end{array}$$
(55)$$\begin{array}{ccc} S_{3}^{\prime }\left(Y|X\right) & \approx & 0.374 \end{array}$$
(56)$$\begin{array}{ccc} T_{3}^{\prime }\left(Y|X\right) & \approx & 0.375 \end{array}$$
□

### Bounds on Conditional Tsallis Entropy

As mentioned in the Introduction, one of the properties of the (conditional) Shannon entropy for discrete variables is to be bounded by the number of elements of the support of the distribution. Furthermore, it is well known that the unconditional Tsallis entropy is always between 0 and $\frac{m^{1-\alpha }}{1-\alpha }$, where *m* is the number of elements in the support of the distribution. In this subsection, we derive bounds for the conditional Tsallis entropies based on the number of elements in the support of each distribution.

**Theorem** **10.**
*Let Z=(X,Y) be any joint random vector defined over sets of size m each. Then,*

(57)
$$\begin{array}{c} 0\leq S_{\alpha }\left(Y|X\right)\leq \frac{m^{1-\alpha }}{1-\alpha } \end{array}$$


(58)
$$\begin{array}{c} 0\leq T_{\alpha }^{\prime }\left(Y|X\right)\leq \frac{m^{1-\alpha }}{1-\alpha }. \end{array}$$

*Moreover all of these lower and upper bounds may be reached by suitable probability distributions P(X,Y).*


**Proof.** The Inequalities (Equation 57) follow from the fact that $S_{\alpha }\left(Y|X\right)$ is the expectation of the unconditional Tsallis entropy.For Inequalities (Equation 58), recall that Equation (Equation 10) can be written, for α<1, as Equation (Equation 12). Note that, for all *x*, the values $T_{\alpha }\left(Y|X=x\right)$ are the (unconditional) Tsallis entropies of the marginal distribution, and are all defined in a set of cardinality *m*.So, by definition of $T_{\alpha }^{\prime }$, for some particular *x*, we have $T_{\alpha }^{\prime }\left(Y|X\right)=T_{\alpha }\left(Y|X=x\right)$. The case α>1 is similar. So, independently of α, for every probability distributions P(X) and P(Y) defined over set with *m* elements, we have $0\leq T_{\alpha }^{\prime }\left(Y|X\right)\leq \frac{m^{1-\alpha }}{1-\alpha }$, since the same bound applies for the unconditional version or any its marginal distributions. □

**Theorem** **11.**
*Let Z=(X,Y) be any joint random vector defined over sets of size m each. Then,*

(59)
$$\begin{array}{ccc} & & \;if\;\alpha >1:\;\;0\leq T_{\alpha }\left(Y|X\right)\leq \frac{m^{1-\alpha }}{1-\alpha }. \end{array}$$

*For α<1, in general, the inequality does not hold.*


**Proof.** Consider first the case α>1. The result follows directly from Inequalities (Equation 15) and (Equation 57).In order to prove that the inequality does not hold for all α<1, consider α=0.1 and the following joint probability distribution:
$$\begin{array}{cccc} X=x\backslash Y=y & 1 & 2 & 3\\ 1 & 1/9 & 1/9 & 0\\ 2 & 1/9 & 1/9 & 1/9\\ 3 & 0 & 1/9 & 1/3 \end{array}$$
Notice that $\frac{m^{1-\alpha }}{1-\alpha }\approx 2.987$ and $T_{0.1}\left(Y|X\right)\approx 3.371$. For any other α<1, one can construct similarly a joint probability distribution for which the inequality is also violated. □

**Theorem** **12.**
*Let Z=(X,Y) be any joint random vector defined over sets of size m each. Then,*

(60)
$$\begin{array}{ccc} & & \;if\;\alpha >1:\;\;0\leq S_{\alpha }^{\prime }\left(Y|X\right)\leq \frac{m^{1-\alpha }}{1-\alpha }. \end{array}$$



**Proof.** The result follows directly from the Inequalities (Equation 26) and (Equation 58). □

We conjecture that the above theorem also holds for α<1. For example, the inequality is true for all uniform probability distribution over *n* variables.

We now show that, for any fixed joint probability distribution P(X,Y), three of the forms of conditional Tsallis entropy studied in this paper are non-increasing functions of α. First, we state a simple theorem.

**Lemma** **1.**
*If $f_{1}\left(x\right)$,…, $f_{m}\left(x\right)$ are non-increasing real functions, then the function $\max_{i}\left(f_{i}\left(x\right)\right)$ is also a non-increasing function.*


**Theorem** **13.**
*For every probability distribution P(X,Y),*
*1.* 
*$T_{\alpha }\left(Y|X\right)$ is a non-increasing function of α.*
*2.* 
*$S_{\alpha }\left(Y|X\right)$ is a non-increasing function of α.*
*3.* 
*$T_{\alpha }^{\prime }\left(Y|X\right)$ is a non-increasing function of α.*



**Proof.** 1.First consider the case α>1, and consider the function $dT_{\alpha }\left(Y|X\right)$, the derivative of the function $T_{\alpha }\left(Y|X\right)$ in order to α:
$$\frac{dT_{\alpha }\left(Y|X\right)}{d\alpha }=\frac{-1+\sum_{x}P{\left(X=x\right)}^{\alpha }}{{\left(\alpha -1\right)}^{2}}-\frac{\sum_{x}\left(\alpha P{\left(Y=y|X=x\right)}^{\alpha -1}\log\alpha \right)}{\alpha -1}.$$
It is easy to see that, since α>1, $\frac{dT_{\alpha }\left(X\right)}{d\alpha }<0$. Therefore, the function $T_{\alpha }\left(Y|X\right)$ is a non-increasing function of α.
Consider now the case α<1 and assume that $\alpha ,\alpha^{\prime }$ are such that $\alpha <\alpha^{\prime }<1$. In order to prove that $T_{\alpha }\left(Y|X\right)$ is non-increasing we have to show that $T_{\alpha }\left(X\right)\geq T_{\alpha }^{\prime }\left(X\right)$, i.e.,:
$$\begin{array}{cccc} & 1-\frac{1-\sum_{x}P{\left(X=x\right)}^{\alpha }}{\alpha -1} & \geq & 1-\frac{1-\sum_{x}P{\left(X=x\right)}^{\alpha^{\prime }}}{\alpha^{\prime }-1}\\ \Leftrightarrow & \frac{-1+\sum_{x}P{\left(X=x\right)}^{\alpha }}{1-\alpha } & \geq & \frac{-1+\sum_{x}P{\left(X=x\right)}^{\alpha^{\prime }}}{1-\alpha^{\prime }}\\ \Leftrightarrow & \frac{1-\sum_{x}P{\left(X=x\right)}^{\alpha }}{1-\alpha } & \leq & \frac{1-\sum_{x}P{\left(X=x\right)}^{\alpha^{\prime }}}{1-\alpha^{\prime }} \end{array}$$
Notice that, since $\alpha <\alpha^{\prime }<1$, Then $\frac{1}{1-\alpha }<\frac{1}{1-\alpha^{\prime }}$ and, therefore, $1-\sum_{x}P{\left(X=x\right)}^{\alpha }\leq 1-\sum_{x}P{\left(X=x\right)}^{\alpha^{\prime }}$. So, the last inequality is true.
2.This part of the result follows from the fact that $S_{\alpha }\left(Y|X\right)$ is the expectation of unconditional Tsallis entropies; see Equation (Equation 6).3.Suppose that α>1. The proof is a direct consequence of Equation (Equation 11) and Lemma 1. The case α<1 can be proven in a similar way.□

It is easy to show that $S^{\prime }$ does not fulfill the property of the last theorem.

**Theorem** **14.**
*There exists probability distributions (X,Y) and $\alpha <\alpha^{\prime }$ for which $S_{\alpha }^{\prime }\left(Y|X\right)\leq S_{\alpha^{\prime }}^{\prime }\left(Y|X\right)$.*


**Proof.** Consider the following joint probability distribution:
$$\begin{array}{ccc} X=x\backslash Y=y & 1 & 2\\ 1 & 0.45 & 0.45\\ 2 & 0.1 & 0.0 \end{array}$$
We have:
$$S_{0.2}^{\prime }\left(Y|X\right)\approx 0.563$$
$$S_{0.5}^{\prime }\left(Y|X\right)\approx 0.621.$$
□

We developed a small application that, given two probability distributions, computes the values of all conditional Tsallis entropies considered in the paper. The application is self-contained and its use is extremely simple. There are two use case examples that the reader can use in order to try the calculator. The interested reader can find it in the following link: http://gloss.di.fc.ul.pt/tryit/Tsallis (accessed on 28 October 2021).

## 6. Conclusions

In this paper, we studied the definitions for the conditional Tsallis entropy existing in the literature. We also considered a possible alternative definition for it. This new proposal is a natural approach to consider as a possible definition. It defines the conditional value as the maximum value of all marginal distributions. Due to this fact, and similar to what happens with the Rényi entropy, this definition was also analyzed, although it was never considered in the literature before. The relationships between the four definitions, described in this work, are summarized in Figure 1.

As we understand, it would be expectable that a proposal for conditional Tsallis entropy would satisfy the following properties:Chain Rule;Convergence to Shannon entropy as the parameter α tended to 1;Its value would be between 0 and the upper bound of the unconditional version.

In Table 1, we summarize the properties that the four proposals have (we also added the property of being a non-increasing function with α). To conclude, we can say that none of the proposals fulfill all of the properties. The definition $T_{\alpha }\left(Y|X\right)$ is the candidate that fulfills more properties.

For future work, since all definitions focus on possible different aspects of the entropy, it would be important to consider a deeper study in this area and its possible applications, aiming to develop a theory that would emphasize the best proposal for each area, or eventually present an ultimate version for the conditional Tsallis entropy that would satisfy all of the desirable properties.

## Figures and Tables

**Figure 1 entropy-23-01427-f001:**
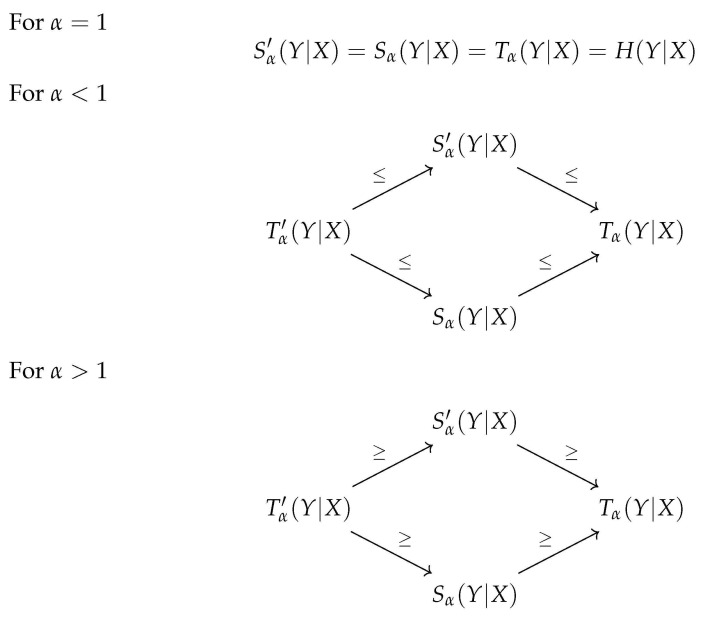
Summary of the relations between the several proposals for the definition of conditional Tsallis entropy.

**Table 1 entropy-23-01427-t001:** Summary of the proved properties of all proposed conditional entropies. The question mark indicates that the property is not known to be fulfilled.

f(Y|X)	$T_{\alpha }\left(Y|X\right)$	$S_{\alpha }\left(Y|X\right)$	$S_{\alpha }^{\prime }\left(Y|X\right)$	$T_{\alpha }^{\prime }\left(Y|X\right)$
Chain Rule	yes	no	no	no
$\lim_{\alpha \to 1}f\left(Y|X\right)=H\left(Y|X\right)$	yes	yes	yes	no
$0\leq f\left(Y|X\right)\leq \frac{{\left|Y\right|}^{1-\alpha }}{1-\alpha }$ and α>1	yes	yes	yes	yes
$0\leq f\left(Y|X\right)\leq \frac{{\left|Y\right|}^{1-\alpha }}{1-\alpha }$ and α<1	no	yes	?	yes
*f* is non-increasing with α	yes	yes	no	yes

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
