# Peer review of "On Conditional Tsallis Entropy"

_entropy, 2021, doi:10.3390/e23111427_

Round 1

Reviewer 1 Report

In this manuscript the notion of conditional Tsallis entropy is studied and one new version of conditional Tsallis entropy is defined and compared with three versions that have been defined in the literature. The manuscript does not provide any clues on which version that may most useful in applications. Actually, it is not clear from the presentation if any of the definitions are preferrable to for instance conditional Rényi entropy. It is also clear that one can concatenate the words "conditional" and "Tsallis entropy" and it is also clear that conditional Shannon entropy is useful for information theory, but as noted in the paper any of the quantities called "conditional Tsallis entropy" in the manuscript will not satisfy all the useful properties that conditional entropy satisfies, so it is not clear if any of the generalizations are useful. These problems should be discussed in some detail in the manuscript.

Details:

Line 10: Tsallis entropy was first defined in [2] and [6]. It is only due to ignorance that the quantity is named after Tsallis.

Line 16-19: In this paragraph is should mentioned that in many (most) of the applications of Tsallis entropy one could have used Rényi entropy instead of Tsallis entropy.

Line 27-28: The notion of mutual information was developed by ideas of communation rather from abstract notions of entropy.

Line 46: Replace "provide the several" by "provide several".

Line 62: Add reference to [2] and [6].

Line 89: The notion of "expected properties" is unclear.

Line 131-132: Unclear, reformulate.

Line 152-153: Unclear, reformulate.

Line 179: Unclear since there is only one uniform distribution.

Line 222: Replace "develop theory" by "develop a theory".

Ref. [1]: Year is missing. 

Reviewer 2 Report

This paper studied three existing definitions of conditional Tsallis entropy and proposed a alternative version. The relation between Tsallis entropy and conditional Tsallis entropy is studied. Some interesting theorems are analyzed. However, its also has some shortcomings. 1. In page 1, line 20. The references about existing conditional Tsallis entropy should be added after “several versions have been proposed and used in the literature”. 2. Some necessary references are indispensable to expand the background. For example, in page 2, line 28 and 29, The latest published work should be introduced about mutual information, symmetry of information and so on. In line30, the application of entropy measure in many fields should be given. 3. In page 8, line 136, “The second Equation” should be corrected to “The second equation”. Remove the similar problems in the paper. 4. In page 10, line 175, “inequalities 26 and 58” should be corrected to “Inequalities (26) and (58)”. 5. In page 10, line 178 and 179, I STRONGLY encourage the author to provide a proof for this conjecture, even a first version proof with problems. The meirt of the quesationable proof has also some suggestive aspects for the other colleagues. 6. In page 11, line 197 to line 200. The proof of Theorem 13 (3) is not precise. The numerical example only illustrates that S’ is not a decreasing function of . However, as a continuous function of α , which is given by author in Theorem 2, increase at first and then decrease or decrease at first and then increase both possible with the condition [ S’(0.2)=0.563, S’(0.5)=0.621]. In other words, It is necessary to prove the Theorem 13 (3) from the point of view of derivatives. Above all, this work needs a minor revision before publication.
